# Lifestyle and Hepatocellular Carcinoma What Is the Evidence and Prevention Recommendations

**DOI:** 10.3390/cancers14010103

**Published:** 2021-12-26

**Authors:** Shira Zelber-Sagi, Mazen Noureddin, Oren Shibolet

**Affiliations:** 1School of Public Health, Faculty of Social Welfare and Health Sciences, University of Haifa, Haifa 3498838, Israel; 2Department of Gastroenterology & Hepatology, Tel Aviv Medical Center, Tel Aviv 6423906, Israel; Orensh@tlvmc.gov.il; 3Karsh Division of Gastroenterology and Hepatology, Cedars Sinai Medical Center, Los Angeles, CA 90048, USA; Mazen.Noureddin@cshs.org; 4Sackler Faculty of Medicine, Tel Aviv University, Tel Aviv 6697801, Israel

**Keywords:** obesity, dietary composition, alcohol, smoking, physical activity

## Abstract

**Simple Summary:**

The increasing public health burden of Hepatocellular carcinoma (HCC) emphasizes the importance of defining important modifiable risk factors. In the following review, we will discuss the evidence for the relation of major lifestyle risk factors, mostly from large population-based studies. Generally, it is has been shown that healthy lifestyle habits, including minimizing obesity, eating a healthy diet, avoidance of smoking and alcohol, and increasing physical activity, have the potential to prevent HCC. Dietary composition is important beyond obesity. Consumption of n-3 polyunsaturated fatty acids, as well as fish and poultry, vegetables and fiber, are inversely associated with HCC, while red meat, saturated fat, cholesterol and sugar are related to increased risk. Data from multiple studies clearly show a beneficial effect for physical activity in reducing the risk of HCC. Smoking and alcohol can lead to liver fibrosis and liver cancer and jointly lead to an even greater risk.

**Abstract:**

The increasing burden of hepatocellular carcinoma (HCC) emphasizes the unmet need for primary prevention. Lifestyle measures appear to be important modifiable risk factors for HCC regardless of its etiology. Lifestyle patterns, as a whole and each component separately, are related to HCC risk. Dietary composition is important beyond obesity. Consumption of n-3 polyunsaturated fatty acids, as well as fish and poultry, are inversely associated with HCC, while red meat, saturated fat, and cholesterol are related to increased risk. Sugar consumption is associated with HCC risk, while fiber and vegetable intake is protective. Data from multiple studies clearly show a beneficial effect for physical activity in reducing the risk of HCC. However, the duration, mode and intensity of physical activity needed are yet to be determined. There is evidence that smoking can lead to liver fibrosis and liver cancer and has a synergistic effect with alcohol drinking. On the other hand, an excessive amount of alcohol by itself has been associated with increased risk of HCC directly (carcinogenic effect) or indirectly (liver fibrosis and cirrhosis progression. Large-scale intervention studies testing the effect of comprehensive lifestyle interventions on HCC prevention among diverse cohorts of liver disease patients are greatly warranted.

## 1. Introduction

Liver cancer remains a global health challenge, and while infection by hepatitis B virus and hepatitis C virus are the main risk factors for hepatocellular carcinoma (HCC) development, non-alcoholic steatohepatitis is associated with metabolic syndrome or diabetes mellitus is becoming a more frequent risk factor [1]. The increasing public health burden of HCC emphasizes the prominent need to define important modifiable risk factors. It is now believed that lifestyle plays a vital role in cancer prevention or progression. Lifestyle includes, among others, physical activity (PA), sedentary behavior, diet and eating habits, all reflected in obesity and abdominal obesity. In addition, smoking and excessive alcohol drinking by themselves or in synergism have been associated with increased risk of HCC. In the following review, we will discuss the evidence for the relation of major lifestyle risk factors with HCC, mostly from extensive population-based studies. Although the medical treatment of HCC is constantly evolving [1,2], primary prevention by lifestyle remains of significant importance, holding the greatest potential of life- and cost-saving.

## 2. Role of Diet and Lifestyle in General in the Prevention of Hepatocellular Carcinoma

Evidence for a potential association between dietary composition and HCC in humans is mainly derived from large observational prospective cohort studies and meta-analyses. Generally, it is shown that the same dietary characteristics and other lifestyle habits (minimizing obesity, smoking, drinking alcohol, and increasing physical activity) that are beneficial in the treatment of non-alcoholic fatty liver disease (NAFLD) also have the potential to prevent HCC, although the HCC-related studies are not specific to NAFLD patients. The association of lifestyle, as a whole, with HCC risk has been tested in a large prospective study by applying a composite score of healthy lifestyle factors consisting of body mass index (BMI), alcohol consumption, cigarette smoking, Mediterranean diet, and sleep duration. After a mean follow-up of 17.7 years, individuals with higher composite scores, representing healthier lifestyles, were at significantly lower risk of HCC in a dose-response manner. A similar inverse association was observed in participants with negative HBsAg and negative hepatitis C virus (HCV)-serology [3]. Similarly, in a meta-analysis of prospective cohort studies looking at a similar healthy lifestyle composite score, adopting the healthiest lifestyles was associated with a 32% lower risk of liver cancer [4].

These studies highlight the importance of comprehensive lifestyle modification strategies for the primary prevention of HCC. In the following review, the specific lifestyle components and their relation to HCC will be summarized.

A summary of all prospective cohort studies and meta-analyses of cohort studies testing the association of diet with HCC is presented (study description and the nutrient/food intake categories which were compared) in Table 1.

## 3. The Role of Obesity in HCC

Obesity has been repeatedly demonstrated to be an independent risk factor for the occurrence of and mortality from primary liver cancer [30,31]. A meta-analysis including 37 prospective studies demonstrated a positive association between overweight and obesity and the risk of liver cancer incidence and liver cancer mortality. The risk for liver cancer incidence or related mortality increased almost two-fold among obese populations, and the relative risks (RR) of liver cancer were 1.32 per each increase of 5 kg/m^2^ BMI [30]. Furthermore, waist circumference is associated with HCC beyond BMI. In fact, adiposity-related markers such as body fat percentage, waist–hip ratio, and waist–height ratio, were all positively related to liver cancer in 437,393 participants from the UK Biobank prospective cohort [32]. In a meta-analysis including five prospective cohort studies, waist circumference was positively associated with the risk of liver cancer in a dose-response manner [33]. Although it seems highly reasonable to recommend weight reduction for the prevention of HCC, according to these prospective studies, and since it is the most evidence-based treatment for NAFLD, a growing etiology for HCC studies that test the direct effect of weight loss on long-term primary prevention are lacking.

## 4. The Role of Dietary Composition beyond BMI

### 4.1. Types of Dietary Fat, Meat, and Fish

Most of the studies dealing with dietary composition and HCC are focused on the fat type. The type of dietary fat, rather than total fat, is related to HCC risk; consumption of n-3 polyunsaturated fatty acids, (PUFA)-rich fish, and individual types of n-3 PUFA, as well as white meat, are inversely associated with HCC, while red meat, saturated fat, and cholesterol are related with increased risk [14,27,28,29]. In a study using data from the Nurses’ Health Study (NHS) and the Health Professionals Follow-up Study (HPFS)**,** after an average follow-up time of 26.6 years, there was no significant association between total fat intake and HCC, but intake of vegetable fats was related with reduced risk of HCC by almost 40% for the highest vs. the lowest quartile. Among fat subtypes, monounsaturated fatty acids (MUFA) and PUFA, including n-3 and n-6 PUFA, were inversely associated with risk of HCC, and higher ratios of MUFA or PUFA to saturated fat were inversely associated with HCC risk [14]. In the same populations of US cohorts, this time focusing on dairy products analysis, intake of high-fat dairy products and butter, but not a low-fat dairy, were associated with a higher risk of incident HCC, implying a role for cholesterol and saturated fat [15]. Consistent findings regarding saturated fat are also noted in the Chinese male population prospective cohort (Shanghai Men’s Health Study), with an average follow-up of 11.91 years. For the highest vs. lowest quartile, saturated fat was related to increased risk by 50%, but surprisingly PUFA was also related to increased risk by 41%, although MUFA was not significantly related to liver cancer [9]. It should be mentioned that in this study, almost 65% of the HCC cases were HBsAg positive; thus, this cohort is less representative of NAFLD-related HCC. Finally, a recent meta-analysis of six prospective cohort studies confirms the robustness of the association with saturated fat; the highest versus lowest intake of dietary saturated fat was associated with a 34% higher liver cancer risk, and this association had a dose-response pattern. There were also significant inverse associations between the ratio of MUFA:saturated fat, unsaturated fatty acids:saturated fat, and liver cancer risk. Cholesterol intake was related to an increased risk of liver cancer by a 16% per increase in 100 mg/day. There were no significant associations between the intake of total dietary fat, MUFA, and PUFA and the risk of liver cancer [34].

Several studies have shown the harmful association between high meat intake and NAFLD [35,36], specifically red and processed meat intake [37,38]. A prospective cohort of the general population from six states in the US with 16-year follow-up data indicated that high intake of total meat, processed and unprocessed red meat (beef, lamb, and pork), and nitrite from processed meat were associated with liver disease-related mortality. In contrast, white meat (poultry and fish) consumption was related to a reduced risk of liver disease-related mortality [39]. Higher saturated fat, cholesterol, iron, and nitrate/nitrite intake are among the possible mechanisms [14,28,29]. In a recent meta-analysis of seven prospective cohort studies, red meat consumption was significantly associated with a greater risk of HCC by 22%, but processed meat consumption was not significantly associated with HCC [40]. However, in an extensive cohort study, processed red meat was associated with almost two times increased risk for HCC (3rd vs. 1st tertile of consumption) [18].

Fish are the main dietary sources of the long-chain omega-3 PUFAs eicosapentaenoic acid (EPA) and docosahexaenoic acid (DHA), and its relation with HCC has been tested in several studies and meta-analyses (see Table 1). An umbrella meta-analysis, incorporating five meta-analyses of prospective cohort studies, demonstrated that every 100 g/day increments in fish consumption were associated with a 35% lower risk of liver cancer [12].

### 4.2. Added (Free) Sugars

The type and quantity of dietary carbohydrates, as quantified by the glycemic index (GI) and glycemic load (GL), and dietary fiber may influence the risk of liver cancer [26]. Among 477,206 participants of the European Prospective Investigation into Cancer and Nutrition cohort (EPIC), higher dietary GI, GL, or an increased intake of total carbohydrate was not associated with liver cancer risk. However, the risk for HCC was increased by 43% per 50 g/day of total sugar, and in contrast, reduced by 30% per 10 g/day of total dietary fiber. The findings for dietary fiber were also confirmed among hepatitis B virus (HBV)/HCV-free participants [26].

One of the primary dietary sources of sugar in a western diet is sugary drinks. In a meta-analysis of observational studies, compared with the lowest level, the highest level of sugar-sweetened beverages (SSB) consumption showed a two-fold increased risk of HCC [41]. However, this was based on only two studies; one is a case-control study among patients with cirrhosis [42]. The other study looked more deeply into the type of soft drinks (sugar- and artificially-sweetened) and fruit and vegetable juices and the risk of HCC, using data from the EPIC cohort from 10 European countries. Combined soft drinks consumption of >6 servings (cans)/week was associated with 80% increased HCC risk vs. non-consumers, adjusting for total energy intake, alcohol consumption, BMI, physical activity, level of educational attainment, and diabetes status. Interestingly, artificially-sweetened soft drinks increased HCC risk by 6% per 1 serving increment. Juice consumption was not associated with HCC risk [43].

## 5. Role of Micronutrients, Fruits, and Vegetables in HCC

There is little research in terms of micronutrients and HCC. In the NIH-American Association of Retired Persons (NIH-AARP) prospective cohort, a higher magnesium intake was independently associated with a lower risk of liver cancer, with intakes in the highest compared with the lowest quartile associated with 35% lower risk. The inverse association was more pronounced in moderate and heavy alcohol users [6]. Similar findings were demonstrated in another prospective follow-up within a screening trial (in 10 screening centers throughout the USA). Total (diet + supplements) magnesium intake was found to be inversely associated with risks of primary liver cancer incidence (about 50% risk reduction) and related mortality (about 40% risk reduction). Similar results were obtained for dietary magnesium intake without supplements (60% risk reduction for intake of >358 mg/day versus intake of <256 mg/day), adjusting for BMI and other lifestyle risk factors [13].

The associations of vegetable and fruit intake with liver cancer risk have been inconsistent in epidemiological studies. In a meta-analysis of observational studies (19 studies were included, ten cohort studies and nine case-control studies), the intake of vegetable, but not fruit, was associated with a lower risk for HCC, which decreased by 8% for every 100 g/day increase in vegetable intake [23]. A later meta-analysis of only 9 prospective cohort studies showed that a higher vegetable intake was associated with a reduction in liver cancer risk, and again, the dose-response analysis indicated that the risk of liver cancer was reduced by 4% per 100 g per day increment of vegetable intake. However, in subgroup analysis, it appeared that the reduction of liver cancer risk was only significant in men. There was also no significant association between fruit intake and liver cancer risk [17].

## 6. Coffee and Tea

Epidemiological studies, mostly prospective cohorts and meta-analyses, repeatedly suggested a protective effect of coffee from HCC [25,44]. In a US Multi-ethnic prospective cohort, compared with non-coffee drinkers, those who drank 2–3 cups per day had a 38% reduction in risk for HCC, and those who drank ≥4 cups per day had a 41% reduction in HCC risk. Furthermore, compared with non-coffee drinkers, participants who consumed 2–3 cups of coffee per day had a 46% reduction in risk of death from chronic liver disease, and those who drank ≥4 cups per day had a 71% reduction. The inverse associations were significant regardless of the participants’ ethnicity, sex, BMI, smoking status, alcohol intake, or diabetes status [44].

One of the recent studies investigated the associations of coffee consumption, including decaffeinated, instant, and ground coffee, with chronic liver disease (CLD) outcomes using the UK Biobank database with electronic linkage to hospital, death, and cancer records. Compared to non-coffee drinkers, coffee drinkers had a 21% lower adjusted risk of CLD and 49% lower risk of death from CLD, but there was no significant evidence for a lower risk of HCC. The associations for decaffeinated, instant and ground coffee individually were similar to all types of coffee combined [45].

Tea is one of the most widely consumed beverages worldwide. In an umbrella Meta-analyses of observational studies, high compared with low green tea consumption was associated with a lower risk of liver cancer by 13%, but the authors conclude that more well-designed prospective studies are needed with consideration of the causes of bias [16].

## 7. Dietary Patterns

Adherence to dietary recommendations, summarized as healthy dietary patterns, has been linked to a reduced risk of developing HCC and dying of CLD [7,19,46,47]. In a case-control study, the Mediterranean diet pattern was associated with lower odds for liver cancer [46]. Furthermore, a prospective study with 32 years of follow-up demonstrated that better adherence to the Alternative Healthy Eating Index-2010, which consists of a high intake of fruit, vegetables, whole grains, nuts and legumes, n-3 fats, and low intake of SSB and fruit juice, red and processed meat, trans fat, sodium, and moderate alcohol consumption, may decrease the risk of developing HCC [19]. Similarly, in the Multiethnic Cohort in the US, among four diet quality index (DQI) scores (Healthy Eating Index-2010, Alternative Healthy Eating Index-2010, Alternate Mediterranean Diet, and Dietary Approaches to Stop Hypertension), only a higher adherence to an alternate Mediterranean diet was associated with a lower risk of HCC. However, all DQI measures were inversely associated with lower CLD mortality. For both outcomes, there was no significant heterogeneity by race/ethnicity [47].

This was also confirmed among a Chinese population, where better adherence to three DQI scores: the Alternative Health Eating Index-2010, Alternate Mediterranean Diet, and Dietary Approaches to Stop Hypertension was related to a lower risk for HCC. Similar inverse associations were observed among HBsAg-negative individuals, and the strongest inverse association was for the alternate Mediterranean diet score [7].

Chronic low-grade systemic inflammation plays an important role in primary liver cancer etiology and can be influenced by dietary patterns. For example, in a cross-sectional study, the dietary inflammatory index (DII) score was demonstrated to be associated with increased levels of various serum inflammatory markers among European adolescents [48]. The DII is composed of 45 food parameters, including micronutrients, macronutrients, bioactive components, or individual foods that were associated with inflammatory biomarkers. A novel prospective large study has investigated the association of DII with primary liver cancer incidence and mortality. Higher DII scores were associated with a two-fold higher risk of developing primary liver cancer and with related mortality, adjusting for other lifestyle risk factors, educational level, BMI, energy intake, and diabetes [11].

In support of these findings, in a cohort study (NHS and HPFS), higher adherence to dietary pattern scores which reflect the dietary inflammatory potential (predictive of inflammatory biomarkers) and patterns related with insulin resistance/hyperinsulinemia (predictive of C-peptide as an indicator of insulin secretion and the triglyceride to high-density lipid-cholesterol ratio as an indicator of insulin resistance), were related with higher risks of developing HCC [8].

Low-carbohydrate diets (LCDs) draw a lot of attention in the treatment of obesity and NAFLD, but little is known about their relationship with the development of HCC. A prospective cohort study (NHS and HPFS) tested this association with two specific subtypes of LCDs; plant-based (determined according to percentages of energy from carbohydrate, plant protein, and plant fat) and animal-based (determined according to percentages of energy from carbohydrate, animal protein, and animal fat) LCDs. Overall, LCD was not associated with HCC risk. Specifically, there was no association between the animal-based LCD score and risk of HCC; however, the plant-based LCD score was inversely associated with HCC risk (adjusted for BMI and other lifestyle parameters). Looking at specific macronutrients, carbohydrate intake, especially from refined grains, was positively associated with HCC, while no association was seen with carbohydrates from whole grains. Animal fat was associated with increased risk, while plant fat was associated with reduced HCC risk [5].

There is less research on the role of healthy eating patterns in the prognosis of HCC. In a cohort of 887 patients with newly-diagnosed, previously untreated HCC survivors, a dietary quality reflected by the Chinese Healthy Eating Index was related with reduced all-cause and HCC-specific mortality, importantly adjusting for age at diagnosis, energy intake, BMI, smoking, education level, alcohol, alpha feto protein (AFP), Child–Pugh class, disease stage, and treatment [10]. These findings suggest that improving diet quality may be useful for the primary prevention of HCC as well as to protect against HCC-related mortality, but the latter needs further confirmation.

## 8. Role of Physical Activity in HCC

The assessment of the impact of PA on cancer risk has been hampered by the fact that there are limited ways to measure PA accurately. Most studies were based on the self-reporting of the amount of PA. A study that assessed the amount of PA “objectively” via accelerometers found a significant discrepancy between the reported and actual PA [49]. Another issue is the definition, length, amount, and intensity of PA, ranging from vigorous aerobic exercise to resistance training or daily activity, which is not sedentary. The topic of PA and HCC was covered in an excellent review by Saran et al. [50], and recently, the role of PA in the prevention of NAFLD-associated HCC was eluded to in a review by Lange et al. [51]. In the following paragraph, we will focus specifically and extensively on the effect of PA on HCC.

### 8.1. Physical Activity and General Cancer Risk

Several studies assessed the association of PA with cancer incidence. In a prospective study covering 1.44 million participants, the authors found that high-level PA was associated with a decrease in 13 types of cancer (top-ranking reduction esophageal, liver, lung, and kidney). An adjustment to BMI mildly attenuated the results, but PA remained associated with a marked reduction in 10 cancers regardless of the adjustment. Surprisingly, PA was associated with an increased risk for melanoma and prostate cancer [52]. A recent review cites five prospective studies assessing the effect of PA on the incidence of cancer; four studies found a protective effect of PA against several digestive, gynecological, and pancreatic cancers, while one study did not find a protective effect of PA against prostate cancer [53]. Data from the NHS suggests that PA during adolescence is associated with a marked decrease in the incidence of breast cancer [54].

### 8.2. Physical Activity and HCC in Animal and Cellular Models

Data on the anti-HCC effect of exercise has been slowly accumulating from animal studies. Aguiar et al. assessed the effect of training on hepatic carcinogenesis. Male Wistar rats were submitted to a DEN-induced liver carcinogenesis protocol and divided into groups according to a high or low-fat diet and swimming training (5 days a week for eight weeks). There was a marked improvement in the biochemical parameters of the trained rats on the low-fat diet (LFD) and high-fat diet (HFD) groups. However, while there was a marked decrease in pre-neoplastic lesion development in the LFD group, no such effect was noticed in the HFD group. The researchers concluded that exercise attenuated liver carcinogenesis together with dietary manipulations [55]. In another study, serum was collected from a young overweight man without metabolic syndrome before and after 3 weeks of LFD and moderate aerobic exercise. Post-intervention serum was added to the HCC cell line HepG-2 and markedly attenuated cell proliferation, lipid accumulation, and signalling of various stress pathways [56]. Both previous studies combined diet and PA as interventions. Indeed, dietary and metabolic signals are known to markedly increase the incidence of HCC. In 2015, Piguet et al. used genetically engineered mice to assess dietary and exercise effects separately. They used hepatocyte-specific PTEN-deficient mice, which develop steatohepatitis and HCC spontaneously. Mice were fed the same diet and were divided into exercise (treadmill running-1 h, five days a week for 32 weeks) or sedentary groups. The exercise group developed hepatic nodules larger than 15 mm^3^ in 71% vs. 100% in the sedentary group. There was also a marked decrease in the number of tumors per liver and tumor volume per liver independently of hepatic steatosis. Exercise was associated with increased AMPK and Raptor phosphorylation and decreased Mammalian Target of Rapamycin (mTOR) activity [57]. Later on, Arfianti A et al. assessed the same effect in a chemically induced DEN HCC model in two groups of genetically engineered mice, obese/diabetic Alms1 mutant (foz/foz) mice, and JNK1 deficient foz/foz mice. Mice were divided to exercise or sedentary groups (exercise wheel from week 4 to 12 or 24 weeks). Exercising, foz/foz mice developed obesity by week 24 but still had less dysplastic hepatocytes and significantly fewer tumors (15% vs. 64% compared to sedentary controls). In contrast to previous studies, these diabetic/obese mice failed to activate AMPK and mTOR Complex 1. Instead, exercise improved insulin sensitivity and steatosis and regulated key signalling pathways, which resulted in decreased hepatocyte proliferation [58]. In another study, tumors derived from an HCC cell line (Morris Hepatoma 3924A) were grown sub-cutaneously and were later resected and surgically implanted in the livers of American cancer Institute rats. The rats were assigned to exercise (treadmill running 1 h, five days a week for four weeks), sedentary or sorafenib± exercise or sorafenib+metformin groups. Tumor area, cell proliferation, and vascular density were all decreased by exercise. AMPK phosphorylation was increased in the exercise group together with the expression of PTEN, while STAT3 and S6 ribosomal proteins were decreased. Transcriptomic analysis showed that exercise affected non-tumoral tissue rather than the tumor itself. The anti-tumoral effects of exercise in the sorafenib group were similar to the effect of metformin, suggesting that metformin induces an exercise-like effect [59]. In a similar study, mice on a choline deficient high-fat diet (CD-HFD) were divided to exercise or sedentary groups. CD-HFD mice developed steatohepatitis and hepatic pre-neoplastic lesions. Exercise improved their metabolic parameters and improved steatohepatitis and hepatic inflammation as manifested by decreased aminotransferases. Similar to the previous study, there were fewer hepatic adenomas with increased AMPK activity and mTOR inhibition. The authors concluded that exercise reduced the transition from fatty liver to non-alcoholic steatohepatitis (NASH) and decreased progression to fibrosis and tumorigenesis [60]. In two recent studies, the intensity and method of exercise were assessed. HCC was induced by DEN in C57BL/6 mice, which were subjected to high-intensity interval training (HIIT) or endurance training on a treadmill (from 8–26 weeks). Endurance training resulted in lower cancer incidence and growth and less fibrosis. Furthermore, endurance training resulted in less lipotoxicity and improved body composition, inflammation, and metabolomics compared to the HIIT group. The authors suggested that moderate-intensity endurance training may be superior to HIIT in its anti-tumoral effect [61]. Similarly, Cao et al. assessed moderate endurance training compared to HIIT, arguing that HIIT may induce an acidic micro-environment that may be tumor-promoting. The DEN induced HCC model was utilized in C57BL/6 mice that were subjected to HIIT or moderate endurance training on a treadmill for 18 weeks. Again, endurance training decreased tumor incidence and size as compared to HIIT. There were no significant differences in the mRNA levels of key gluconeogenesis enzymes [62].

The data presented from mouse and cellular models, although preliminary, show a beneficial effect of exercise on HCC initiation and progression in chemically, metabolically, and surgically implanted HCC induced models. While these models do not bear a good resemblance to human HCC, the data is compelling. Furthermore, recent, small publications suggest that exercise type is important, as endurance training showed better anti-tumoral effects than HIIT. Finally, several of the models suggest a synergistic effect to a healthy diet on tumor incidence in these models.

### 8.3. Physical Activity and HCC Incidence and Prevention in Humans

Initial data on the effect of PA on HCC risk in humans came from prospective studies in Korea and Japan. In Korea, a prospective study of 444,963 men older than 40 showed that moderate-high leisure-time physical activity (LPA) (>2 times/week, >30 min) had a significant protective effect against several cancers, including HCC as compared to low LPA (<2 per week, <30 min) [63]. In Japan, 79,771 men and women aged 45–74 responded to a questionnaire and were followed until 2004. LPA was assessed using a metabolic equivalents/day score. Increased PA was associated with a decrease in the risk for various cancers and was more pronounced in women, especially among the elderly and those who exercised regularly. HCC was decreased only in men and reached significance only in the group who were in the highest quartile of PA compared to the lowest [64]. Wen et al. developed prediction models to assess the risk of HCC in a large cohort in Taiwan. Their cohort consisted of 428,584 individuals that were divided into two groups, one with HCV (130,533) and the other without. Data were collected from a standard medical screening program, and the average follow-up was 8.5 years. During this period, 1668 HCC cases were identified. The data included PA, which was categorized as inactive, low-active, and active according to the intensity in a metabolically equivalent task (MET); X represented the duration of exercise in hours per week (METs hours/week). Although individuals with low active and active PA showed significantly reduced risk for HCC in the initial model, this difference became non-significant when other variables (age, sex, health history, HBV and HCV, AFP and transaminases) were added to the model [65]. A study conducted in the USA in 2013 focused on 507,897 participants aged 50–71 in the NIH-AARP diet and health study that were followed for ten years. A total of 628 cases of HCC were identified during the study period. Physical exercise was defined as the performance of 20 min of vigorous activity per week. Groups were classified between “rarely perform” (lowest level) to >5 times per week (highest level). Comparison of the lowest to highest group of PA showed a significant decrease in the risk for HCC (RR = 0.64; 95% CI, 0.49–0.84) independently of BMI [66]. Arem et al. looked at PA in almost 300,000 men and women in the same NIH-AARP study in order to assess PA patterns over the life course and their association with HCC. They used modelling starting from teenage years to middle age and identified seven distinct PA trajectories. They showed that those who maintained PA levels over life had an approximately 30% reduction in HCC risk compared to those with consistently low PA. The specific pattern of PA (increased or decreased PA through life) had different evolving risks of HCC. Their results suggest that maintaining PA from early age onwards had the best protective effect and warn that increasing PA later in life may not yield the same protective effect [67].

Recent years have shown a plethora of studies assessing the role of PA in association with hepato-pancreato-biliary (HPB) cancers. Data from the EPIC cohort looked at cause-specific hazard ratios (HR) among 467,000 participants focusing on HPB cancers. They identified 275 HCCs among 532 all HPB cancers during follow-up. There was a 45% decrease in the risk for HCC comparing active to inactive participants and a 50% decrease for those engaging in vigorous PA (defined a >2 h/week). Markers associated with obesity such as BMI and waist circumference partially explained the reduced risk [68]. In a rigorous prospective study spanning almost 30 years, the authors assessed the effect of various lifestyle parameters, including PA, on liver-related mortality in over 125,000 participants. HCC accounted for a third of the deaths. The risk for overall liver-related mortality declined progressively with increasing PA and increased with higher BMI. The HR for liver-related mortality tripled in obese sedentary compared to lean, active participants. Findings were similar for HCC or cirrhosis-specific mortality. The authors suggested that engaging in average pace walking >3 h/week could have prevented 25% of liver-related deaths [69]. More recently, the emphasis focused on the intensity, mode, and duration of PA. Luo et al. looked at moderate intensity PA in two well-defined cohorts of the NHS and the HPFS over an average of 23 years. Surprisingly, total and vigorous PA was not associated with a reduced risk of HCC, while moderate-intensity PA showed an inverse association with HCC. The reduced risk was especially associated with brisk walking, suggesting that the mode of PA may play a role in HCC prevention [70]. Lee et al. wanted to establish whether there is a minimum PA threshold for the prevention of HCC. They conducted a meta-analysis and divided the PA performed in the selected studies into three groups: high >3 h/week, moderate 2–3 h/week, or low <2 h/week. A total of 10 prospective cohort studies were included. PA was associated with a dose-dependent decrease in HCC risk and mortality. High and moderate PA reduced the risk of HCC by 54% and 45%, respectively. According to their data, the authors state that 2 h/week PA is mandatory to reduce HCC risk [71]. An interesting recent study looked at variability in the association of obesity and PA and HCC at the state level in the US. Trends of HCC incidence from 2001–2017 were calculated using data from the Centers for Disease Control (CDC) and the National Cancer Institute Surveillance. There were striking state-level disparities in HCC incidence ranging from 6.3 to 0.9 in various states and ethnicities. There was a moderate inverse correlation with state-level PA and the incidence of HCC (r = −0.40, *p* = 0.004) [72].

Two studies looked at the effect of PA on specific, at-risk populations. In the first, Feng et al. looked at the effect of PA on alcohol-related cancer (including liver cancer). Data were collected from British and Scottish population-based surveys spanning the years 1994–2008. Alcohol consumption was categorized from “never drinkers” to “harmful” (>35 units/week for women; >49 units/week for men) and PA to lower (7.5 METs h/week) or upper recommended limits (15 METs h/week). There were 54,686 participants, and hazardous/harmful alcohol consumption was associated with a marked increase in cancer-related mortality. Although cancers were not separated by type, the increased risk was eliminated among participants who exercised more than 7.5 METs hours/week and persisted in the upper recommended limits group [73].

Lastly, a recent study looked at whether PA is associated with HCC risk in patients with chronic HBV infection. The authors looked at 9727 treatment-naïve Korean HBV carriers who started treatment with nucleoside/tide analogous from 2012–2017. During the study period, the cumulative HCC incidence was 8.3%. There was an inverse correlation between carriers engaged in PA measured in METs and those without PA. PA had a protective effect in patients with and without cirrhosis that resulted in an approximately 40% reduction in risk for HCC. The authors conclude that PA was significantly associated with a reduced risk of HCC in HBV carriers, treated with anti-viral medications [74].

The data presented here from multiple studies, cohorts, and meta-analyses, clearly shows a beneficial effect for PA in reducing the risk of HCC. The data comes from diverse populations with different gender and ethnic backgrounds. However, there are multiple questions remaining to be answered, especially the obesity independent effect of PA that was only partially shown. Other issues include the duration, mode, and intensity of PA and whether PA is effective in reducing the risk for HCC across multiple liver diseases.

### 8.4. Physical Activity Following HCC Treatment in Humans

Following the promising data on HCC prevention prior to tumor development, researchers set out to assess whether PA can improve recovery, prevent a recurrence, and possibly prolong survival in patients with HCC that were treated with or without curative intent.

Kiabori et al. from Japan, looked at 51 patients who underwent hepatectomy for HCC. The patients were randomized to diet alone or diet and PA after the operation. PA was started one week before the operation, resumed one week post-operatively, and continued for six months. The authors show that patients in the PA group had significantly improved metabolic parameters such as whole-body mass, fat mas, insulin resistance and insulin levels and recommend early resumption of PA after HCC resection [75]. The same group assessed whether perioperative PA was associated with long-term survival in a larger group of HCC patients undergoing hepatectomy. One hundred and six patients underwent cardiopulmonary exercise assessment utilizing various methods pre and six months post-hepatectomy. Patients were classified as the maintenance group if they had >90% anaerobic threshold six months post-operatively compared to pre-operatively (*n* = 78) or a decreased group if the threshold was below 90% (*n* = 28). 5-year recurrence-free survival and overall survival were significantly improved in the maintenance group compared to the decreased group (39.9% vs. 9.9% *p* = 0.018, and 81.9% vs. 61.7% *p* = 0.006 respectively). Thus the maintenance of the anaerobic threshold, which is maintained by PA, was an independent positive prognostic marker [76].

Some concern was raised, whether PA could exacerbate CLD in patients with HCC. It was thought that exercise might increase portal pressures and decrease glomerular filtration rate and thus may expose cirrhotic patients to the risk of variceal bleeding and the development of hepatorenal syndrome.

Koya et al. observed that patients with HCC spent most of their time in bed during hospitalization for treatment; therefore, they engaged them in PA during hospitalization and assessed 6-min walking distance, deterioration in liver function, and loss of muscle mass. They enrolled 54 patients with CLD and HCC (median age 76 years) in a light exercise program during hospitalization (2.5–4 METs/20 min/day). There was no worsening of liver function, and six min-walk was maintained at discharge. The addition of branch-chain amino acid (BCAA) minimized muscle atrophy. The authors conclude that PA and BCAA are important during hospitalization in patients with CLD and HCC and do not pose a risk for CLD deterioration for these elderly patients [77]. The same group tested their results in patients with HCC undergoing Transarterial Chemoembolization (TACE). They enrolled 209 HCC patients who underwent TACE and randomized them to an exercise group (*n* = 102) or control (*n* = 107). Patients were engaged in a light exercise program during hospitalization, and the effect on muscle mass was assessed by changes in skeletal muscle index (∆SMI). Although there were no changes in basic lab parameters between the groups, there was a marked increase in ∆SMI in the exercise group vs. controls (+0.28 cm^2^/m^2^ vs. −1.11 cm^2^/m^2^, *p* = 0.0029 respectively). The authors state that in-hospital exercise in patients with HCC undergoing TACE increased muscle mass and prevented sarcopenia [78]. Narao et al., another Japanese group, looked at whether in-hospital PA improved activities of daily life (ADL) in patients with HCC hospitalized for treatment. Nineteen patients were enrolled with a median age of 78 years—85% who had stage II HCC. During hospitalization, they performed a wide range of aerobic and resistance training and stretching for 20–60 min/day. There was no change in Child–Pugh class before and after exercise. ADL was assessed by the functional independence measure (FIM), which is a combined activity score. The FIM score markedly increased after PA (*p* = 0.0156), and especially the stairs index (5.9 vs. 6.4 points *p* = 0.024 before and after PA respectively). Thus PA improved muscle strength and ADL without worsening CLD [79]. In 2021, a multicenter study was conducted in Japan to assess the effect of in-hospital exercise on frailty in HCC patients. Patients were classified into exercise and non-exercise groups. The exercise group was treated with light-moderate aerobic and resistance training (20–40 min/day, median four days). The liver frailty index (LFI) was used to assess frailty. During hospitalization, the LFI significantly improved in the PA compared to the non-exercise group (∆LFI -0.17 vs. −0.02 *p* = 0.012). Exercise and being female were identified as independent factors for improving LFI. The authors suggest that in-hospital exercise may be beneficial in improving physical function in patients with HCC [80].

As seen, data on the effect of PA following treatment of HCC on “hard” endpoints such as overall survival, progression-free survival, and recurrence is entirely lacking. There is only limited data on “soft” endpoints such as frailty, ADL, and sarcopenia. However, given that PA did not worsen the underlying liver disease, it seems logical to recommend PA following the treatment of HCC.

## 9. Role of Alcohol in HCC

### 9.1. Alcohol as a Carcinogen 

Ethanol is first metabolized by alcohol dehydrogenase (ADH) into acetaldehyde which enters the mitochondria and becomes oxidized via acetaldehyde dehydrogenase (ALDH). Upon excessive alcohol intake, the endoplasmic reticulum or peroxisomes metabolize the extra alcohol via the cytochrome p4502E1 (CYP2E1) and reactive oxidative species (ROS) generate causing cell damage by forming DNA and protein adducts. CYP2E1 also triggers the formation of acetaldehyde. Oxidative stress secondary to ROS accumulation damages cell components through lipid peroxidation and DNA mutagenesis via the formation of adducts and the impairment of repair mechanisms. Acetaldehyde also alters methyl transfer, leading to DNA hypomethylation associated with modifications to gene expression (oncogenes and tumor suppressor genes). Alcohol leads to the suppression of MAT1a, the main enzyme responsible for the synthesis of the main methyl donor of the liver, S-adenosylmethionine. The MAT1a knockout mouse develops steatohepatitis, cirrhosis, and HCC. The microenvironment surrounding HCC shifts from CD8+ T cells to tumor-associated macrophages and M2 macrophages. Ethanol consumption also leads to NK-cell dysfunction, which plays a role in tumor surveillance. Finally, patients with PNPLA3 (rs 738409) polymorphism have been linked to HCC in patients with alcoholic liver disease. Other studies have linked the TM6SF2 polymorphism to alcoholic-related HCC.

Due to the lack of alcohol-induced HCC models, the mechanism by which ethanol can cause HCC is incompletely understood. Nevertheless, it is believed that the cause is related to the carcinogenic properties of acetaldehyde (alcohol metabolite), oxidative stress induced by ethanol, abnormal DNA methylation, alcohol-induced tumor microenvironment, and immune system-related changes.

Once ethanol is absorbed in the small intestine and transported to the liver, it is metabolized in the cytoplasm of hepatocytes by alcohol dehydrogenase (ADH) to acetaldehyde, which enters the mitochondria to be oxidized to acetaldehyde dehydrogenase (ALDH) [81]. In excessive alcohol use, the mitochondria may become exhausted, and the endoplasmic reticulum or peroxisomes metabolize the extra alcohol via the cytochrome p4502E1 (CYP2E1) [81]; this process often leads to the formation of reactive oxidative species (ROS). Some human ALDH isoenzymes, such as ALDH2, are associated with HCC development [82]. On the other hand, acetaldehyde itself forms DNA adducts and protein adducts to stimulate various kinds of collagen, such as type I collagen, and thus liver fibrosis [83,84]. All these processes contribute to the carcinogenic properties of acetaldehyde. Regarding ROS, they are formed after excessive alcohol (~>40 g/day) consumption, resulting in alcohol metabolism via CYP2E1. The formation of ROS leads to cariogenic signals via various mechanisms, including a) formation of lipid peroxidation products, such as malondialdehyde and 4-hydroxy-2-nonenal, which cause a mutation of the P53 gene (commonly found in HCC) [85,86], and b) mediation of tumor angiogenesis and metastasis via NF-kb, VEGF, and MCP-1 [87]. In addition to ROS, iron intake from the intestine increases with excessive alcohol intake, leading to iron overload in the liver, which has been associated with DNA strand breaks and P53 mutation [88].

The aberrant methylation of DNA or proteins leads to the formation of cancers, including HCC. Alcohol leads to suppression of MAT1a, the main enzyme responsible for the synthesis of the main methyl donor of the liver, S-adenosylmethionine [89]. The MAT1a knockout mouse develops steatohepatitis, cirrhosis, and HCC [89]. The microenvironment surrounding the tumor also is very important. In that regard, research in HCC animal models has revealed a shift in the microenvironment surrounding HCC from CD8+ T cells to tumor-associated macrophages and M2 macrophages [90,91]. The immune system plays a role in the cross-talk between alcohol and HCC formation. Alcohol consumption leads to NK-cell dysfunction, which plays a role in tumor surveillance. NK cells in patients with alcoholic cirrhosis have diminished cytotoxic activity against cancers cells [92,93]. In addition, Toll-like receptor 4 (TLR4) identifies lipopolysaccharides (LPS) (products of gram-negative bacteria) that are usually elevated in persons with excessive alcohol intake and alcoholic cirrhosis [94]. TLR4 and the LPS pathway have been linked to HCC progression [94,95]. Finally, genetic associations between HCC development and alcoholic liver disease have been identified: patients with PNPLA3 (rs 738409) polymorphism have been linked to HCC in patients with alcoholic liver disease [96]. Other studies have linked the TM6SF2 polymorphism to alcoholic-related HCC, but further research is needed to confirm the genetic role of alcohol in HCC [97]. The Pathways involved in alcohol-mediated liver carcinogenesis are summarized in Figure 1. 

### 9.2. The Amount of Alcohol as a Risk Factor

The amount of alcohol consumed has been linked to cirrhosis, and by itself, is a risk of HCC development. Daily alcohol consumption of 30–50 g is a risk of cirrhosis development [98,99], while more than 60–100 g/day are needed for HCC development [100,101]. More than 280 million people worldwide meet the definition of alcohol use disorder. It is thought that 10–20% of those who drink heavily will develop cirrhosis, and the incidence of alcohol-induced cirrhosis is around 1.9–2.6% [102,103]. The incidence of HCC in alcoholic cirrhosis is not firmly known; however, data from a few cohorts show an annual incidence of 2.1–5.6% (Table 2).

### 9.3. Gender, Obesity, and Type 2 Diabetes as Additional Synergistic Risk Factors with Alcohol

Females may develop cirrhosis with less alcohol consumption (20 g/day) than males (60–80 g/day) over the same time (~10 years) [98,108]. Females are also at a 5-fold higher risk of developing HCC, with 80 g/day consumption of alcohol in comparison with the same consumption in men [109]. This difference could be due to the low levels of ADH in the stomach of women, which leads to higher systemic levels of alcohol and thus greater exposure of the liver to alcohol [110]. The role of estrogen has been proposed as a contributing factor; however, data in this regard are conflicting [110]. Estrogen could increase Kupffer cell response to LPS, which can be associated with TLR4 activation and more severe inflammatory response and cellular injury [111]. On the other hand, research in animal models has found that estrogen may reduce the risk of HCC via an interleukin-6 inhibitory mechanism [112,113]. The conflicting data call for further investigation into the role of gender differences and alcohol consumption in HCC development.

Type 2 diabetes increases the risk for HCC. In a population-based study, subjects with a history of type 2 diabetes had an odds ratio (OR) of 2.7 (95% CI, 1.6–4.3) for HCC development compared with those who did not have type 2 diabetes. The authors found a synergistic interaction between HCC risk among heavy alcohol consumption and type 2 diabetes (OR = 4.2; 95% CI, 2.6–5.8) [114].

The mechanism by which type 2 diabetes increases the risk of HCC is unknown. However, the associated hyperinsulinemia may induce liver inflammation via the release of proinflammatory cytokines such as TNF-2 (alpha), IL6, and NF-KB. Hyperinsulinemia and type 2 diabetes have also been associated with steatosis and steatohepatitis. In addition, the increase in ROS can lead to oncogenic signals, as described earlier, and P53 mutation. Finally, free fatty acid production resulting from hyperinsulinemia may activate c-Jun amino-terminal kinase 1 (JNK1) that stimulates cellular proliferation and suppression of apoptosis. The activation of the insulin growth factor (IGF) pathway was found to play a role in the pathogenesis of HCC [115]. IGF1 leads to phosphorylation of insulin receptor substrate 1, which activates the AKT/mTOR and mitogen-activated protein kinase pathways that inhibit apoptosis and stimulate cell proliferation.

Similarly, obesity has been associated with HCC. In a prospective study conducted on Taiwanese men, alcohol use and obesity, defined as BMI ≥ 30 kg/m^2^, had synergistic effects: HCC incidence with HR of 3.41 (95% CI, 1.25–9.27; *p* < 0.025) [116]. The authors stratified subjects per the World Health Organization categories of BMI and found that, with alcohol use, the risk of incident HCC in overweight increases, with HR of 2.4 (95% CI, 1.3–4.4); in obesity, with HR of 2.0 (95% CI, 1.1–3.7); and in morbidly obese with HR of 2.9 (95% CI, 1.0–8.0). The underlying mechanisms by which obesity contributes to increased risk of HCC in addition to ethanol is likely similar to that in type 2 diabetes.

### 9.4. Viral Hepatitis and Alcohol-Associated HCC

The coexistence of viral hepatitis and excessive alcohol intake increases the risk of HCC. However, with anti-viral therapies leading to the cure of chronic HCV and control of chronic HBV, this association has become less concerning. Patients with alcohol-associated HCC, compared with those who have HCV-associated HCC, have delays in surveillance, and thus they present at more advanced stages of HCC, which are less amenable to curative interventions and have worse outcomes [117].

### 9.5. Change of HCC Risk after Alcohol Cessation

The degree of decline in risk of HCC after alcohol cessation is unclear. A meta-analysis from 4 studies suggested a decline of 6–7% per year; however, the authors themselves cautioned about uncertainty around the interpretation of the results; they projected a period of 23 years of drinking cessation to equal that of a never drinker [118].

## 10. Role of Smoking in HCC

Smoking and alcohol consumption are strongly associated with each other. Thus, smoking is considered a confounding factor in liver diseases and HCC, but its role is not established. There is evidence, though, that smoking can lead to liver fibrosis and liver cancer.

In a Europe-wide cohort, former smoking contributed to almost half of HCC cases (47.6%), whereas HBV and HCV infection contributed 13.2% and 20.9%, respectively [119]. In that study, former smokers and current smokers had an OR of 1.98 for developing HCC, while heavy alcohol use had an OR of 1.77. In a meta-analysis from 38 cohort studies and 58 case-control studies, the adjusted meta-RR of liver cancer was 1.51 (95% CI, 1.37–1.67) for current smokers and 1.12 (95% CI, 0.78–1.60) for former smokers compared with the risk in non-smokers [120]. In a study of 104 male patients with HCC and CLD who were compared to 104 males without HCC and chronic liver disease, the authors found that the RR for developing HCC was 17.9 among those with both alcohol drinking and smoking [121]. This risk was higher than in those who practiced one of these habits individually. This risk decreased in previous smokers if they did not drink or quit drinking, though the RR remained high, 9.4. In a large study from the US (the Liver Cancer Pooling Project), which is a consortium of US-based prospective cohort studies that included data from 1,518,741 individuals and had HCC and intrahepatic cholangiocarcinoma (ICC) cases, current smokers had an increased risk of HCC HR: 1.86 (95% CI, 1.57–2.20) and ICC HR : 1.47 (95% CI, 1.07–2.02) [122]. Interestingly, in individuals who quit smoking more than thirty years ago, the risk of HCC risk was equivalent to the risk in those who never smoked HR : 1.09 (95% CI, 0.74–1.61).

Smoking may assert its carcinogenic effects via direct or indirect oncogenic effects. The indirect effect may occur via the toxic substances of smoking products, which may activate IL6, IL8, and TNF-a, which is an oxidative stress that can lead to activation of stellate cells [123,124,125]. Stellate cell activation leads to fibrosis progression and eventually contributes to the effects of alcohol on the liver, thus accelerating cirrhosis. Another indirect mechanism is that of decreased oxygen-carrying capacity effect of tissues, which increases erythropoietin concentration and secondary iron absorption from the small intestine to the liver. This action leads to iron overload in the liver, oxidative stress, and hepatocellular injury. The direct effect of smoking may occur via the carcinogens found in smoking products, such as tar and vinyl chloride, leading to effects on tumor suppressing genes and P53 [123,126,127]. Cigarettes are rich in 4-aminobiphenyl, which has been associated with HCC [128]. N-Nitrosodimethylamine is a product of cigarettes and leads to liver fibrosis and HCC. Cadmium, which is present in cigarettes, has been strongly associated with HCC [129]. Lastly, it has been shown that hepatic monoamine oxidase B (MAOB) is involved in the biosynthesis of geranylgeranoic acid (GGA) (found in animal models and hepatoma cell lines) which is believed to prevent hepatocarcinogenesis [130,131]. Nevertheless, smoking tobacco inhibits MABOB, which can lead to suppressing GGA and its preventive effect of hepatocarcinogenesis [132]. More studies are needed to confirm the role of these factors and smoking in HCC.

## 11. Conclusions

The evidence for the role of lifestyle in HCC is primarily observational but based upon high-quality prospective studies that allow temporal inference, with large samples sizes, robust methods, and consistent results. Multiple prospective cohort studies drive the evidence for primary prevention. It is stronger than the evidence for tertiary prevention, which needs to be further studied with “hard” endpoints such as overall survival, progression-free survival and recurrence. One of the limitations of these observational studies is that diet, physical activity, alcohol, and smoking are reported, leading to some information bias. There is still a gap in knowledge on the practical amount and type of physical activity, alcohol, and nutrients that should be recommended to prevent liver cancer. Large-scale randomized trials testing the effect of lifestyle interventions on HCC prevention among diverse cohorts of liver disease patients are warranted to provide evidence-based prevention guidelines. However, such studies are challenging to perform due to the low incidence rate of HCC, requiring an immense sample size and long-term follow-up to collect a sufficient number of cases that will provide evidence for intervention effects. Therefore, we need to rely on observational and pre-clinical studies and implement healthy lifestyle behaviors, which have embedded benefits for preventing other morbidities. Preventive interventions, combining personal medical and nutritional advice with public health strategies such as education, increasing awareness among physicians and the public, and policy measures to make the environment friendlier to healthy lifestyle practices, could have an enormous positive impact. A practical summary of comprehensive lifestyle behaviors related to HCC prevention is depicted in Figure 2.

## Figures and Tables

**Figure 1 cancers-14-00103-f001:**
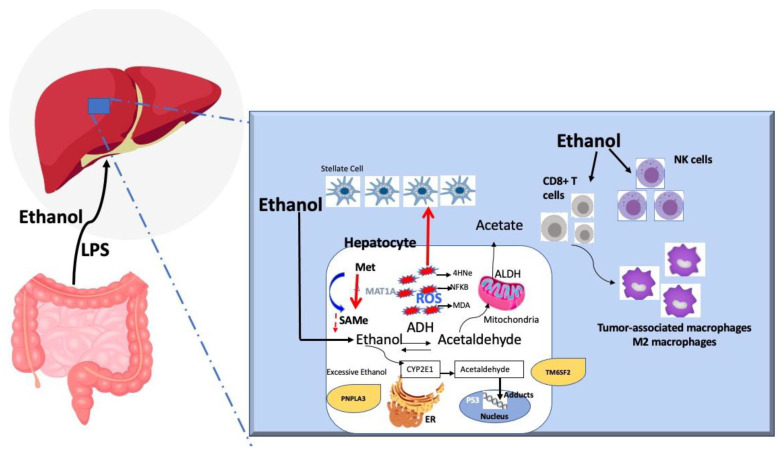
Pathways involved in alcohol-mediated liver carcinogenesis.

**Figure 2 cancers-14-00103-f002:**
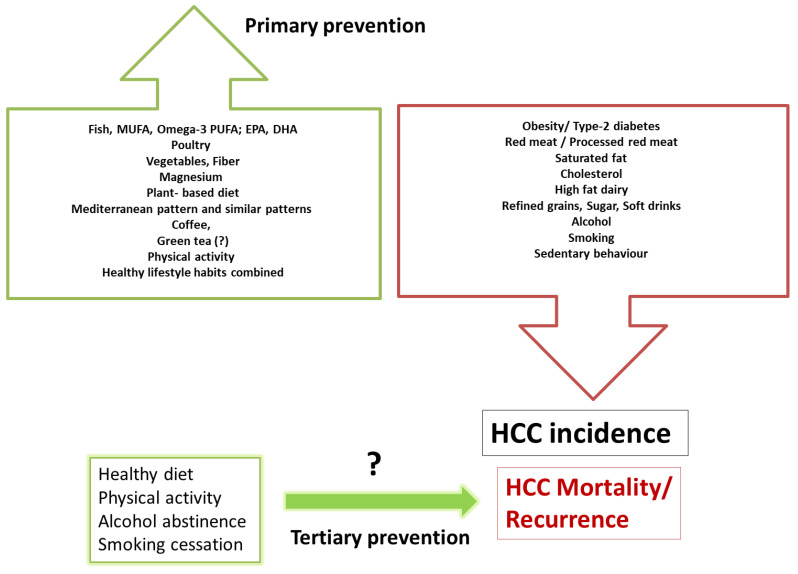
Practical summary for prevention by lifestyle habits; behaviors related with increased risk or reduced risk for HCC incidence and outcomes. The evidence for primary prevention is driven from many prospective cohort studies and seems to be more evidence-based than the scarce evidence for tertiary prevention.

**Table 1 cancers-14-00103-t001:** Prospective cohort studies and meta-analyses of cohort studies testing the association between dietary factors and patterns and hepatocellular carcinoma.

Author, Year of Publication (Ref)	Study Design Cohort Study/Meta-Analysis of Cohort Studies	Study Population and Sample Size	Nutrient/Food Group	Adjusted HR/RR (CI) of Highest Category vs. Lowest Category	Nutrient/Food Intake Categories Which Were Compared (Highest Category vs. Reference Category)
Liu Y., 2021 [5]	Prospective cohort	Nurses’ Health Study (*n* = 88,770 women). The Health Professionals Follow-up Study (*n* = 48,197 men)	Plant based low-carbohydrate diet	0.83 (0.70–0.98)	Per 1 standard deviation increase
			Carbohydrates from refined grains	1.18 (1.00–1.39)	Per 1 standard deviation increase
			Plant fat	0.78 (0.65–0.95)	Per 1 standard deviation increase
Shah SC., 2021 [6]	Prospective cohort	The NIH-American Association of Retired Persons (NIH-AARP) Diet and Health Study (*n* = 536,359)	Magnesium (diet + supplements)	0.65 (0.48–0.87)	4th vs. 1st quartile
Luu HN., 2021 [7]	Prospective cohort	Singapore Chinese Health Study (*n* = 63,2570)	Alternative Health Eating Index-2010 (AHEI-2010)	0.69 (0.53–0.89)	4th vs. 1st quartile
			Alternate Mediterranean Diet (aMED)	0.70 (0.52–0.95)	4th vs. 1st quartile
			Dietary Approaches to Stop Hypertension (DASH)	0.67 (0.51–0.87)	4th vs. 1st quartile
Yang W., 2021 [8]	Prospective cohort	Nurses’ Health Study (*n* =70,055 women). Health Professionals Follow-up Study (*n* = 49,261 men)	Empirical lifestyle pattern score for hyperinsulinemia (ELIH)	1.89 (1.25–2.87)	3rd vs. 1st tertile
			Empirical lifestyle pattern score for insulin resistance (ELIR)	2.05 (1.34–3.14)	3rd vs. 1st tertile
			Empirical dietary inflammatory pattern (EDIP)	2.03 (1.31–3.16)	3rd vs. 1st tertile
Ji XW., 2021 [9]	Prospective cohort	Chinese men (*n* = 59 998)	Total fat	1.33 (1.01–1.75)	4th vs. 1st quartile
			Saturated fat	1.50 (1.13–1.97)	4th vs. 1st quartile
			Monounsaturated fat	1.26 (0.96–1.65)	4th vs. 1st quartile
			Polyunsaturated fat	1.41 (1.07–1.86)	4th vs. 1st quartile
Luo Y., 2020 [10]	Prospective cohort	Patients with new HCC enrolled in the Guangdong Liver Cancer Cohort (*n* = 887)	Chinese Healthy Eating Index (CHEI-2016)	0.74 (0.56–0.98) Outcome: HCC specific mortality	3rd vs. 1st tertile
			Healthy Eating Index-2015 (HEI-2015)	0.93 (0.71–1.21) Outcome: HCC specific mortality	3rd vs. 1st tertile
Zhong GC., 2020 [11]	Prospective cohort	American adults from the prostate, lung, colorectal and ovarian cancer screening trial (*n* = 103,902)	Dietary inflammatory index (DII) from food and supplements	2.05 (1.23–3.41) Outcome: PLC incidence	3rd vs. 1st tertile
			Dietary inflammatory index (DII) from food and supplements	1.97 (1.13–3.41) Outcome: PLC mortality (*n* = 102)	3rd vs. 1st tertile
			Dietary inflammatory index (DII) from food only	2.57 (1.44–4.60) Outcome: PLC incidence	3rd vs. 1st tertile
Jayedi A., 2020 [12]	Umbrella Review of Meta-Analyses of Prospective Cohort Studies (5 Meta-analyses)	Mixed populations	Fish	0.65 (0.48–0.87)	per 100 gr/day
Zhong GC., 2020 [13]	Prospective cohort	American adults from the prostate, lung, colorectal and ovarian cancer screening trial (*n* = 104,025)	Magnesium (diet + supplements)	0.44 (0.24–0.80) Outcome: PLC incidence	3rd vs. 1st tertile
			Magnesium (diet + supplements)	0.83 (0.67–1.01) Outcome: PLC incidence	Per 100 mg/d
			Dietary magnesium	0.41 (0.22–0.76) Outcome: PLC incidence	3rd vs. 1st tertile
			Dietary magnesium	0.65 (0.51–0.82) Outcome: PLC incidence	Per 100 mg/d
			Magnesium (diet + supplements)	0.37 (0.19–0.71) Outcome: PLC mortality	3rd vs. 1st tertile
Yang W., 2020 [14]	Prospective cohort	Nurses’ Health Study (*n* =88,657 women). Health Professionals Follow-up Study (*n* = 49,826 men)	Vegetable fats	0.61 (0.39–0.96)	17.7 vs. 8.7 (% energy)
			n-3 PUFA	0.63 (0.41–0.96)	0.8 vs. 0.5 (% energy)
			n-6 PUFA	0.54 (0.34–0.86)	6.5 vs. 3.7 (% energy)
Yang W., 2020 [15]	Prospective cohort	Nurses’ Health Study (*n* = 93,427 women). Health Professionals Follow-up Study (*n* = 51,418 men)	High-fat dairy	1.81 (1.19–2.76)	2.0 vs. 0.4 serving/day
			Low-fat dairy	1.18 (0.78, 1.78)	1.9 vs. 0.2 serving/day
			Butter	1.58 (1.06–2.36)	0.7 vs. 0 serving/day
			Yogurt	0.72 (0.49–1.05)	0.2 vs. 0 serving/day
Kim TL., 2020 [16]	Umbrella Review of Meta-analyses of observational studies (2)	Mixed populations	Green tea	0.87 (0.78–0.98)	High vs. low
Guo XF., 2019 [17]	Meta-analysis (9 cohorts)	1,326,176 participants	Vegetable	0.96 (0.95–0.97)	Per 100 gr/d
Ma Y., 2019 [18]	Prospective cohort	Nurses’ Health Study (*n* = 92,389 women). Health Professionals Follow-up Study (*n* = 50,468 men).	Processed red meat	1.84 (1.16–2.92)	3rd vs. 1st tertile
			Total white meat	0.61 (0.40–0.91)	3rd vs. 1st tertile
			Unprocessed red meat	1.06 (0.68–1.63)	3rd vs. 1st tertile
			Poultry	0.60 (0.40–0.90)	3rd vs. 1st tertile
			Fish	0.70 (0.47–1.05)	3rd vs. 1st tertile
Ma Y., 2019 [19]	Prospective cohort	Nurses’ Health Study (*n* = 121,700 women). Health Professionals Follow-up Study (*n* = 51,529 men)	Alternative Healthy Eating Index-2010 (AHEI-2010)	0.61 (0.39–0.95)	3rd vs. 1st tertile
Tran KT., (2019) [20]	Prospective cohort	UK Biobank population (*n* = 471,779)	Coffee	0.50 (0.29–0.87)	Any consumption vs. none
			Instant coffee	0.51 (0.28–0.93)	Any consumption vs. none
			Ground coffee	0.47 (0.20–1.08)	Any consumption vs. none
Kennedy OJ., 2017 [21]	Meta-analysis (18 cohorts)	Mixed populations, 2,272,642 participants	Coffee	0.71 (0.65–0.77)	An extra two cups per day
	2 cohorts	Approximately 850,000 participants	Caffeinated coffee	0.73 (0.63–0.85)	An extra two cups per day
	3 cohorts	Approximately 750,000 participants	Decaffeinated coffee	0.86 (0.74–1.00)	An extra two cups per day
Gao M., 2015 [22]	Meta-analysis (3 cohorts)	Mixed populations, 693,274 participants	Fish	0.73 (0.56–0.90)	Highest vs. lowest consumption
Yang Y., 2014 [23]	Meta-analysis (9 cohorts)	Mixed populations, 1,474,309 participants	Vegetables	0.66 (0.51–0.86)	Highest vs. lowest consumption
Luo J., 2014 [24]	Meta-analysis (7 cohorts)	Mixed populations, 2,677,514 participants	Red meat	1.43 (1.08–1.90)	Highest vs. lowest consumption
			White meat	0.70 (0.57–0.86)	Highest vs. lowest consumption
			Fish	0.74 (0.61–0.91)	Highest vs. lowest consumption
Bravi F., 2013, [25]	Meta-analysis (8 cohorts)	Mixed populations, 378,392 participants	Coffee	0.64 (0.52–0.7)	No consumption vs. any consumption
Fedirko V., 2013 [26]	Cohort	European Prospective Investigation into Cancer and Nutrition cohort (*n* = 477,206)	Total sugar	1.43 (1.17–1.74)	Per 50 gr/day
			Total dietary fiber	0.70 (0.52–0.93)	Per 10 gr/day
Sawada N., 2012 [27]	Prospective cohort	Population-based prospective cohort of Japanese subjects (*n* = 90,296)	Fish (rich in n-3 PUFA)	0.64 (0.42–0.96)	70.6 vs. 9.6 gr/day
			EPA	0.56 (0.36–0.85)	0.74 vs. 0.14 g/day
			DHA	0.56 (0.35–0.87)	1.19 vs. 0.28 g/day
Freedman ND., 2010 [28]	Cohort	Men and women of the National Institutes of Health–AARP Diet and Health Study (*n* = 495,006)	White meat	0.52 (0.36–0.77)	65.8 vs. 9.7 g/1000 kcal
			Red meat	1.74 (1.16–2.61)	64.8 vs. 10 g/1000 kcal
Ioannou GN., 2009 [29]	Cohort	General US population from the first National Health and Nutrition Examination Survey (*n* = 9221)	Cholesterol	2.45 (1.3–4.7)	≥511 vs. <156 mg/d

**Table 2 cancers-14-00103-t002:** The incidence of HCC in alcoholic cirrhosis.

Study	Number	Location	Length of Follow Up (Year)	HCC Cases (#)	Incidence
Torisu et al. [104]	47	Japan	6.8	9	2.1
Kodama et al. [105]	85	Japan	3.0	6	2.5
Mancebo et al. [103]	450	Spain	3.5	62	2.6
N’kontchou et al. [106]	478	France	4.2	108	5.6
Ganne-Carrie et al. [107]	652	France/Belgium	2.4	43	2.9

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
