# Peer review of "Lifestyle and Hepatocellular Carcinoma What Is the Evidence and Prevention Recommendations"

_cancers, 2021, doi:10.3390/cancers14010103_

Round 1

Reviewer 1 Report

Dear authors,

You have submitted quite a fascinating manuscript and I would like to see it printed future. I have few objections to the manuscript. However, some issues need to be addressed.

Comment 1.

    I feel that the description of smoking is little unsatisfactory compared to the fascinating description of diet, physical activity, and alcohol consumption. As a recent study of smoking and liver cancer, I suggest some papers.

These papers were described inhibition of MAOB by compound(s) contained in cigarette smoke and the biosynthesis of lipids that suppress hepatocarcinogenesis by monoamine oxidase type B in the liver. 

The inhibition of monoamine oxidase type B by compound(s) contained in cigarette smoke. 
PMID: 25808895, 12829416

About the biosynthesis of lipids that suppress hepatocarcinogenesis by monoamine oxidase type B in the liver.
PMID: 32094232, 34564450

    Please, consider quoting the above literature. And/or, I hope that there will be more descriptions about the relationship between smoking and HCC, not limited to the above paper.

Comment 2.

Where is Figure 2 mentioned in the text?
If necessary, add "conclusion" section and refer to the content of Figure 2 in the text.

Minor point

  1. The styles of citations are not unified.

      Please check everything again and correct.
      (Volume, issue, page ..........)
      For examples Reference No.14, 17, 22 and more.

  1. Delete unnecessary space and underbar.

     L98.  including n-3  and -> including n-3 and

     L284. Physical activity and HCC in animal and cellular models_

              →Similarly for L353, L446

I hope these comments will be helpful.

Best regards

Author Response

Dear authors,

You have submitted quite a fascinating manuscript and I would like to see it printed future. I have few objections to the manuscript. However, some issues need to be addressed.

Response: We are very grateful for this kind feedback.

Comment 1.

    I feel that the description of smoking is little unsatisfactory compared to the fascinating description of diet, physical activity, and alcohol consumption. As a recent study of smoking and liver cancer, I suggest some papers.

These papers were described inhibition of MAOB by compound(s) contained in cigarette smoke and the biosynthesis of lipids that suppress hepatocarcinogenesis by monoamine oxidase type B in the liver. 

The inhibition of monoamine oxidase type B by compound(s) contained in cigarette smoke. 
PMID: 25808895, 12829416

About the biosynthesis of lipids that suppress hepatocarcinogenesis by monoamine oxidase type B in the liver.
PMID: 32094232, 34564450

    Please, consider quoting the above literature. And/or, I hope that there will be more descriptions about the relationship between smoking and HCC, not limited to the above paper.

 Response: These suggestions have been added to the end of the smoking paragraph in a concise way as MAOB is not quite proven to play a role in HCC. As mentioned at the beginning of the paragraph the role of smoking in HCC is not quite established.

Comment 2.

Where is Figure 2 mentioned in the text?
If necessary, add "conclusion" section and refer to the content of Figure 2 in the text.

 Response: Thank you very much for this correction. A reference to Figure 2 was now added at the end of the manuscript.

Minor point

  1. The styles of citations are not unified.

      Please check everything again and correct.
      (Volume, issue, page ..........)
      For examples Reference No.14, 17, 22 and more.

  1. Delete unnecessary space and underbar.

     L98.  including n-3  and -> including n-3 and

     L284. Physical activity and HCC in animal and cellular models_

              →Similarly for L353, L446

Response: This has been corrected per Endnote style. We realize that still some DOIs are missing, in case it could not be corrected by ENDNOTE, we will make sure to add it once the manuscript is approved in a final version.

Thank you very much for the corrections; all were incorporated.

Reviewer 2 Report

This is a comprehensive review on the association of lifestyle and the occurrence of hepatocellular carcinoma. The manuscript was well prepared. In this article, the authors presented an overview regarding the influence of lifestyle on the occurrence of HCC, and recommended how to prevent HCC by changing lifestyle. This review provided useful information for the clinicians to manage the patients with potential risk(s) of HCC.

Author Response

This is a comprehensive review on the association of lifestyle and the occurrence of hepatocellular carcinoma. The manuscript was well prepared. In this article, the authors presented an overview regarding the influence of lifestyle on the occurrence of HCC, and recommended how to prevent HCC by changing lifestyle. This review provided useful information for the clinicians to manage the patients with potential risk(s) of HCC.

Response:

Response: We are very grateful for these positive comments.

Reviewer 3 Report

Dear Editor, thank you for inviting me to revise this manuscript.

Unfortunately, although the topic is interesting and extremely timely, we believe the paper is unsuitable for publication in the journal due to several issues.

The work lacks originality, and does not provide additional information regarding this topic compared to similar papers. In fact, in the past months several other reviews were published on the same topic, so this review lacks originality – in my view.

Author Response

Dear Editor, thank you for inviting me to revise this manuscript.

Unfortunately, although the topic is interesting and extremely timely, we believe the paper is unsuitable for publication in the journal due to several issues.

The work lacks originality, and does not provide additional information regarding this topic compared to similar papers. In fact, in the past months several other reviews were published on the same topic, so this review lacks originality – in my view.

Response: We are familiar with previous recent reviews on this topic; however, we believe this suggested review is very updated and comprehensive in all four lifestyle topics it covers and therefore has an added value.   

Round 2

Reviewer 1 Report

The manuscript has been revised well.
Please correct typo etc... before publication.

Author Response

Thank you, we rechecked the manuscript for typoes. 

Reviewer 3 Report

Dear Editor, thank you so much for inviting me to revise this manuscript about HCC.

This study addresses a current topic.

The manuscript is quite well written and organized. English could be improved.

Figures and tables are comprehensive and clear.

The introduction explains in a clear and coherent manner the background of this study.

We suggest the following modifications:

  • Introduction section: although the authors correctly included important papers in this setting, we believe a couple of studies should be cited within the introduction (PMID: 33479224 ; PMID: 34167433), only for a matter of consistency. We think it might be useful to introduce the topic of this interesting study.
  • Methods and Statistical Analysis: nothing to add.
  • The authors should expand some sections, including a more personal perspective to reflect on. For example, they could answer the following questions – in order to facilitate the understanding of this complex topic to readers: what potential does this study hold? What are the knowledge gaps and how do researchers tackle them? How do you see this area unfolding in the next 5 years? We think it would be extremely interesting for the readers.

However, we think the authors should be acknowledged for their work. In fact, they correctly addressed an important topic , the methods sound good and their discussion is well balanced.

One additional little flaw: the authors could better explain the limitations of their work, in the last part of the Discussion.

We believe this article is suitable for publication in the journal although major revisions are needed. The main strengths of this paper are that it addresses an interesting and very timely question and provides a clear answer, with some limitations.

We suggest a linguistic revision and the addition of some references for a matter of consistency. Moreover, the authors should better clarify some points.

Author Response

We thank the reviewer for his thoughtful comments. The introduction part has been extended, and the suggested references were added. In the last part of the discussion, we added the knowledge gaps, studies weaknesses, and how we see this area evolving. Some of the authors are native English speakers, and they read the manuscript thoroughly again to correct typos.

Round 3

Reviewer 3 Report

The authors modified the manuscript according to our suggestions.

We recommend Acceptance.